# PI-GNN: Towards Robust Semi-Supervised Node Classification against Noisy Labels

## Abstract

Semi-supervised node classification, as a fundamental problem in graph learning, leverages unlabeled nodes along with a small portion of labeled nodes for training. Existing methods rely heavily on high-quality labels, which, however, are expensive to obtain in real-world applications since certain noises are inevitably involved during the labeling process. It hence poses an unavoidable challenge for the learning algorithm to generalize well. In this paper, we propose a novel robust learning objective dubbed *pairwise interactions (PI)* for the model, such as Graph Neural Network (GNN) to combat noisy labels. Unlike classic robust training approaches that operate on the *pointwise interactions* between node and class label pairs, PI explicitly forces the embeddings for node pairs that hold a positive PI label to be close to each other, which can be applied to both labeled and unlabeled nodes. We design several instantiations for PI labels based on the graph structure and the node class labels, and further propose a new uncertainty-aware training technique to mitigate the negative effect of the sub-optimal PI labels. Extensive experiments on different datasets and GNN architectures demonstrate the effectiveness of PI, yielding a promising improvement over the state-of-the-art methods.

## 1 Introduction

Graphs are ubiquitously used to represent data in different fields, including social networks, bioinformatics, recommendation systems, and computer network security. Accordingly, graph analysis tasks, such as node classification, have a significant impact in reality (Lan et al., 2020). The success of machine learning models, such as graph neural networks (GNNs) on node classification relies heavily on the collection of large datasets with human-annotated labels (Zhou et al., 2019). However, it is extremely expensive and time-consuming to label millions of nodes with high-quality annotations. Therefore, when dealing with large graphs, usually a subset of nodes is labeled, and a wide spectrum of semi-supervised learning techniques have emerged for improving node classification performance (Zhu et al., 2003; Zhou et al., 2003; Kipf & Welling, 2017).

Although achieving promising results, these techniques overlook the existence of noisy node labels. For instance, practitioners often leverage inexpensive alternatives for annotation, such as combining human and machine-generated label (Hu et al., 2020), which inevitably yields samples with noisy labels. Since neural networks (including GNNs) are able to memorize any given (random) labels (NT et al., 2019; Zhang et al., 2017), these noisy labels would easily prevent them from generalizing well. Therefore, training robust GNNs for *semi-supervised node classification against noisy labels* becomes increasingly crucial for safety-critical graph analysis, such as predicting the identity groups of users in social networks or the function of proteins to facilitate wet laboratory experiments, etc.

A natural solution is extending the robust training techniques specifically designed for image data into graphs, such as by estimating noise transition matrix (NT et al., 2019), re-weighting training samples (Xia et al., 2021; Li et al., 2021; Dong et al., 2020) and modifying model structure (Li et al., 2020a), which achieve promising results. However, there are two limitations. Firstly, some techniques require a clean set of node and label pairs for training, which are not practical to obtain under extremely noisy settings (Li et al., 2021). Secondly, all of them solely rely on the noisy *pointwise* input-label interactions during training. Although some of them *implicitly* exploit other attributes (the clean graph structure, etc.) that encodes the *pairwise* information between nodes, such as aggregating features by GNNs on the input graphs or launching label propagation to get the pseudo

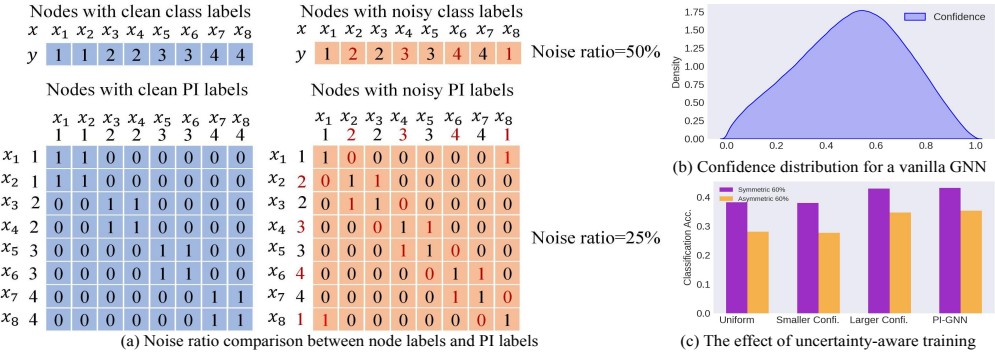

Figure 1: (a) Noise ratio comparison with noisy node labels, which shows the noise ratio of PI labels is much smaller than that of node labels. Red number means noisy label. (b) Density plot for the produced confidence mask with 60% symmetric noise. x-axis denotes the confidence value, where there exist uncertain PI predictions. (c) Test accuracy of the GNN trained by the PI loss from all node pairs (Uniform), the node pairs with larger (Larger Confi.) and smaller confidence (Smaller Confi.), all node pairs weighted by the confidence mask (PI-GNN). Models trained on node pairs with larger confidence perform much better than those trained with lower confidence or without PI labels. PI label is instantiated as the adjacency matrix here.

labels for unlabeled nodes, they still operate on the noisy embeddings for inference, which may lead to limited performance. Thus, designing a robust GNN training technique for node classification against noisy labels is still a challenging problem.

In this paper, we propose a novel framework, which *explicitly* leverages the **P**airwise **I**nteractions in **G**raph **N**eural **N**etwork (**PI-GNN**) to perform robust training without auxiliary clean labels. The PI, which is based on the similarity of two nodes, is meaningful for GNN against noisy labels because the noise rate for the PI labels can be much lower than that of the pointwise noisy class labels (Figure 1(a)). Consider two nodes from the same class have the same noisy labels, their similarity is still high *w.r.t.* the PI, which is helpful for the model to learn. Meanwhile, the pointwise information will inject both of these noisy class labels into GNN, which hurts its generalization performance.

To ensure an effective PI learning procedure, two important components are highlighted. The **first** one is *how to design informative PI labels*. We propose two instantiations: 1) we use the adjacency matrix as PI labels so that if two nodes are connected, their PI label is positive. 2) We compare the noisy labels, i.e., two nodes that have the same class labels are given a positive PI label. During training, PI-GNN explicitly forces the embedding similarity of two nodes to be close to the PI labels. The **second** one is *how to deal with the sub-optimal PI labels during regularization*. It is critical to notice that the two instantiations are not optimal. Firstly, connected nodes do not guarantee to have the same (clean) class label, so their PI label is not always positive. Secondly, the class labels are noisy, and comparing them to obtain the PI labels inevitably introduces noise. To address this problem, we introduce uncertainty estimation, which quantifies the reliability of each PI label (Figure 1(b)). Specifically, we propose to exploit a separate GNN only trained with the PI learning objective and calculate a confidence mask for each node pair from its PI prediction. Then, we filter out uncertain pairs by weighting the PI loss from another GNN jointly trained with the PI loss and node classification loss. Figure 1(c) shows the accuracy of GNNs trained on the node pairs with higher and lower confidence on CiteSeer (Yang et al., 2016) and the former one performs better, which justifies the intuition of PI-GNN. Our main contributions are summarized as follows:

- We propose robust GNNs against noisy labels for node classification, which serve as a crucial step towards the reliable deployment of GNNs in complex real-world applications.
- We introduce a promising strategy to explicitly model the pairwise interactions and an uncertainty estimation approach to mitigate the negative effects of sub-optimal PI labels.
- We demonstrate PI-GNN can be effectively applied on different datasets, GNN architectures and different noise types and rates, e.g., improving the test accuracy by 6.7% on CiteSeer with 80% asymmetric noise.

## 2 RELATED WORK

**Graph neural networks.** Graph neural networks have been widely used to model the graph-structured data with various architectures, such as graph convolutional network (GCN) (Kipf & Welling, 2017), graph attention network (GAT) (Velickovic et al., 2018), GraphSAGE (Hamilton

et al., 2017), Graph Isomorphism Network (GIN) (Xu et al., 2019), Simple Graph Convolution (SGC) (Wu et al., 2019), etc. Common graph analysis tasks, including node classification (Park & Neville, 2019; Oono & Suzuki, 2020), link prediction (Baek et al., 2020; Zhang & Chen, 2018), graph classification (Bacciu et al., 2018; Errica et al., 2020), graph generation (Liao et al., 2019; Shi et al., 2020), have been widely studied in literature. However, only a few works focused on training robust GNNs against noisy labels, such as by loss correction (NT et al., 2019) for graph classification, sample re-weighting (Xia et al., 2021; Li et al., 2021) for node classification. None of them exploited explicit pairwise interactions, which are compared with our PI-GNN in Section 5.4. Bui et al. (2017); Stretcu et al. (2019) utilized graph structures for semi-supervised learning but with clean labels.

**Neural networks with noisy labels.** Methods for neural networks against noisy labels can be roughly categorized into three types (Han et al., 2020b), i.e., approaches from the perspective of data (van Rooyen & Williamson, 2017), objective (Reed et al., 2015; Miyato et al., 2019) and optimization (Arpit et al., 2017). Methods based on data mainly built the noise transition matrix to explore the data relationship between clean and noisy label by an adaptation layer (Sukhbaatar et al., 2015), loss correction (Patrini et al., 2017) and prior knowledge (Han et al., 2018a). Methods based on objective modified the learning objective by regularization (Han et al., 2020a), reweighting (Liu & Tao, 2016; Wang et al., 2017) and loss redesign (Thulasidasan et al., 2019). Methods based on optimization mainly changed the optimization policy, such as by memorization effect (Jiang et al., 2018), self-training (Ren et al., 2018) and co-training (Yu et al., 2019). Wu et al. (2020) proposed to use the similarity loss for noisy labels on image data but it relied on the noisy transition matrix, which is sensitive to the matrix estimation quality and cannot use the graph structure for regularization. In this paper, we extend several approaches from each category to compare with PI-GNN in Section 5.4.

## 3 PRELIMINARY

**Graph Neural Networks.** Let $G = (V, E)$ be a graph with node feature vectors $X_v$ for $v \in V$ and edge set $E$. GNNs use the graph structure and node features $X_v$ to learn a representation vector of a node $h_v$, or the entire graph $h_G$, which usually follow a neighborhood aggregation strategy and iteratively update the representation of a node by aggregating representations of its neighbors. After $k$ iterations of aggregation, a node's representation captures the structural information within its $k$-hop network neighborhood. Formally, the $k$-th layer of a GNN is

$$a_v^{(k)} = \text{AGGREGATE}^{(k)}\left(\left\{h_u^{(k-1)} : u \in \mathcal{N}(v)\right\}\right), \quad h_v^{(k)} = \text{COMBINE}^{(k)}\left(h_v^{(k-1)}, a_v^{(k)}\right), \quad (1)$$

where $h_v^{(k)}$ is the feature vector of node $v$ at the $k$-th layer. $h_v^{(0)} = X_v$. $\mathcal{N}(v)$ denotes the neighboring nodes of $v$. The choices of $\text{AGGREGATE}^{(k)}(\cdot)$ and $\text{COMBINE}^{(k)}(\cdot)$ can be diverse among different GNNs. For example, in GCN, the element-wise mean pooling is used, and the AGGREGATE and COMBINE steps are integrated as follows:

$$h_v^{(k)} = \text{ReLU}\left(W \cdot \text{MEAN}\left\{h_u^{(k-1)}, \forall u \in \mathcal{N}(v) \cup \{v\}\right\}\right), \quad (2)$$

where $W$ is a learnable matrix. For node classification, each node $v \in V$ has an associated label $y_v$, the node representation $h_v^{(K)}$ of the final layer is used for prediction.

**Label-noise representation learning for GNNs.** Let $X_v$ be the feature and $y_v$ be the label for node $v$, we deal with a dataset $\mathcal{D} = \{\overline{\mathcal{D}}^{\text{tr}}, \mathcal{D}^{\text{te}}\}$ which consists of training set $\overline{\mathcal{D}}^{\text{tr}} = \{(A, X_v, \overline{y}_v)\}_{v \in V}$ that is drawn from a corrupted distribution $\overline{D} = p(A, X, \overline{Y})$ where $\overline{Y}$ denotes the corrupted label. Let $p(A, X, Y)$ be the non-corrupted joint probability distribution of features $X$ and labels $y$, and $f^*$ be the (Bayes) optimal hypothesis from $X$ to $y$. To approximate $f^*$, the objective requires a hypothesis space $\mathcal{H}$ of hypotheses $f_\theta(\cdot)$ parametrized by $\theta$. A robust algorithm against noisy labels contains the optimization policy to search through $\mathcal{H}$ in order to find $\theta^*$ that corresponds to the optimal function in the hypothesis for $\overline{\mathcal{D}}^{\text{tr}} : f_{\theta^*} \in \mathcal{H}$, and meanwhile is able to assign correct labels for $\mathcal{D}^{\text{te}}$.

## 4 PROPOSED APPROACH

In this section, we introduce our proposed PI-GNN, which mitigates the negative effects of noisy labels for semi-supervised node classification by explicitly exploiting the pairwise interactions in

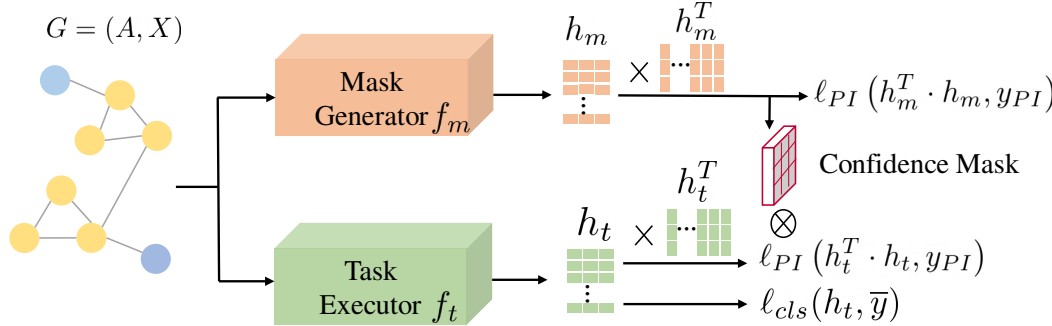

Figure 2: The framework of our PI-GNN, which has two GNNs, i.e., a mask generator and a task executor for robust semi-supervised node classification. The two GNNs $f_m, f_t$ learn the pairwise interactions between each node pair by enforcing the embedding similarity to be close to the PI labels $y_{PI}$. The mask generator generates a confidence mask, which is applied to the PI loss of the task executor to reduce the uncertainty of its predictions caused by the sub-optimal PI labels. $\times$ denotes dot product and $\otimes$ means element-wise multiplication.

**GNNs.** In what follows, we will first provide a method overview and then illustrate the learning objective to enhance the pairwise interactions in PI-GNN (Section 4.1). We introduce the uncertainty-aware robust training approach by using PI for regularization in Section 4.2.

**Overview.** Figure 2 demonstrates the overview of PI-GNN, which is composed of two different GNNs. The first one is only trained with the PI learning objective, whose outputs are used to generate the confidence mask. We denote it as the mask generator $f_m$. The second GNN is trained with both the PI learning objective and the noisy node classification objective, which is used to perform the node classification task. We denote it as the task executor $f_t$. The PI loss for each node pair in the task executor is multiplied by the confidence mask from the mask generator in order to reduce the uncertainty caused by the collected sub-optimal PI labels.

### 4.1 LEARNING FROM PAIRWISE INTERACTIONS

Let us suppose the semantic class labels for certain nodes in the training set $\overline{\mathcal{D}}^{\text{tr}} = \{(A, X_v, \bar{y}_v)\}_{v \in V}$ are corrupted. Since, ultimately, we are interested in finding a GNN model $f$ parametrized by $\theta$ that minimizes the generalization error on a clean test set $\mathcal{D}^{\text{te}}$, a natural solution is to exploit additional information for the learning algorithm to find a robust parameter $\theta^*$ in the hypothesis space $\mathcal{H}$. One straightforward candidate for such information is leveraging the pairwise interactions between two nodes to perform extra regularization, whose learning objective is shown to hold a much smaller noise rate than that with the noisy class labels (Figure 1(a)).

**Construct informative PI labels.** In order to perform PI learning, the first step is to construct reliable PI labels $y_{PI} \in \mathbb{R}^{|V| \times |V|}$ to alleviate the negative effect of noisy labels for the GNNs. Here $|V|$ is the cardinality of the vertex set on the input graph $G$. While a reasonable choice of $y_{PI}$ is by comparing whether two nodes have the same class label $y$ and assign those with the same class label a positive PI label, it is *impossible* to obtain such PI labels with noisy class labels $\bar{y}$. Therefore, assume $i, j \in V$ and $\wedge$ as the "and" operator, we relieve such condition by proposing two relaxed instantiations:

$$y_{PI}^1 = A \quad \text{or} \quad y_{PI}^2 = \bar{y}_i \wedge \bar{y}_j, \tag{3}$$

where the first one models the PI between two nodes by their connectivity in the input adjacency matrix $A$. The second one firstly performs label propagation (Zhu & Ghahramani, 2002) to get the class labels for each node in the input graph. Then it assigns the PI label of node pair $<i, j>$ to 1 if their noisy class labels are the same and 0 otherwise. Both of them are not optimal because firstly, the fact that two nodes are connected does not mean their clean class labels are the same. Secondly, comparing the noisy class labels also creates noisy PI labels if the same clean class labels are corrupted to two different noisy labels despite the noise rate is greatly reduced. We provide an uncertainty-aware training approach in Section 4.2 to alleviate this problem.

**Learning objective for enhancing pairwise interactions.** Given node embeddings $h$ which is calculated by $h = f(A, X, \theta)$, let $h_i^T \cdot h_j$ be the dot product between two embeddings, and $P(h_i^T \cdot h_j)$

be the estimated probability of the node pair $<i,j>$ has the positive pairwise interaction, the PI learning objective $\ell_{PI} \in \mathbb{R}^{|V| \times |V|}$ is formulated as follows:

---

**Algorithm 1** PI-GNN: Learning Pairwise Interactions for GNNs against noisy labels

---

**Input:** Input graph $G = (V, E, X)$ with noisy training data $\overline{\mathcal{D}}^{\text{tr}} = \{(A, X_v, \bar{y}_v)\}_{v \in V}$, randomly initialized GNNs $f_m$ and $f_t$ with parameter $\theta_m$ and $\theta_t$, weight for PI loss $\beta$, pretraining epoch $K$ for $f_m$. Total training epoch $N$.

**Output:** Robust GNN $f_t$ against noisy labels.

**for** $epoch = 0; epoch < N; epoch + +$ **do**

    **if** $epoch \leq K$ **then**

        Set $M = \mathbb{1}$, update the parameter $\theta_m$ of the mask generator $f_m$ by Equation 4 and the parameter $\theta_t$ of the task executor $f_t$ by Equation 7.

    **else**

        Estimate the confidence mask $M$ by Equation 5 with the mask generator $f_m$.

        Calculate the PI loss $\ell_{PI}^t$ for the task executor $f_t$ by Equation 6.

        Update the parameter $\theta_m$ of the mask generator $f_m$ by Equation 4 and the parameter $\theta_t$ of the task executor $f_t$ by Equation 7.

    **end**

**end**

**return** The model $f_t$.

---

$$\ell_{PI}(h; y_{PI}) = \sum_{<i,j> \in B_{PI}^+} -\log P(h_i^T \cdot h_j) + \sum_{<i,j> \in B_{PI}^-} -\log(1 - P(h_i^T \cdot h_j)), \tag{4}$$

where $B_{PI}^+, B_{PI}^-$ denote the node pairs that hold the positive and negative PI labels, respectively.

## 4.2 UNCERTAINTY-AWARE ROBUST TRAINING

**Uncertainty estimation.** In order to train a robust GNN that is not sensitive to the sub-optimal PI labels, we resort to *uncertainty estimation* and reduce the negative effect of the node pairs that the model is uncertain about during regularization (Equation 4). Specifically, we measure the uncertainty by calculating the confidence map of the PI predictions as follows:

$$M(i, j) = \begin{cases} \sigma(h_i^T \cdot h_j), & y_{PI}(i, j) = 1 \\ 1 - \sigma(h_i^T \cdot h_j), & y_{PI}(i, j) = 0 \end{cases} \tag{5}$$

where $y_{PI}(i, j)$ denotes the PI label between node $i$ and $j$ and $\sigma(\cdot)$ is the sigmoid function. The confidence map $M$ measures the uncertainty by looking at the closeness between the prediction and the given PI label. If the prediction becomes close to the given labels more easily, then the reliability of the PI labels is higher and more attention should be paid for such node pairs.

Therefore, we introduce a re-weighting mechanism for the PI loss $\ell_{PI} = \ell_{PI} \otimes M$ where the re-weighted PI loss is obtained by multiplying the confidence mask and its original PI loss in an element-wise way. However, since the GNN is trained with the noisy class labels $\bar{y}$ at the same time, the confidence mask $M$ cannot be estimated well.

**Decoupling with two branches.** In this paper, as shown in Figure 2, we propose to decouple the confidence mask estimation and node classification by using two separate GNNs, which are referred as a mask generator $f_m$ and a task executor $f_t$. The mask generator generates the confidence mask $M$ by only learning with the PI objective. The task executor uses the mask $M$ from the mask generator to re-weight different node pairs in order to combat the noisy PI labels as follows:

$$\ell_{PI}^t = \ell_{PI}^t \otimes M, \tag{6}$$

where $\ell_{PI}^t$ denotes the PI loss for the task executor $f_t$ and $\otimes$ means element-wise multiplication.

The PI learning procedure allows for explicitly exploiting the pairwise interactions between two nodes, resulting in a GNN that is affected less by the noisy class labels. Wu et al. (2020) employed similarity labels $y_{PI}^2$ for regularization. However, it transforms the noise transition matrix estimated for noisy class labels $\bar{y}$ to correct the similarity labels, which is sensitive to the matrix estimation quality. Meanwhile, the pairwise interactions from the input adjacency matrix cannot be explored.

**Overall training.** Put them together, we introduce a new robust training objective for node classification against noisy labels on GNNs, leveraging the pairwise interactions in Section 4.1. The key idea is to perform the node classification task by the task executor $f_t$ while regularizing $f_t$ to produce similar embeddings for nodes that have a closer PI and vice versa. The overall uncertainty-aware training objective for the task executor $f_t$ is formulated as:

$$\ell_t = \ell_{cls}^t(f_t(A, X, \theta_t), \bar{y}) + \beta \cdot \ell_{PI}^t, \tag{7}$$

where $\beta$ is a hyperparameter to balance the node classification loss $\ell_{cls}^t$ and the PI loss $\ell_{PI}^t$. Besides, the mask generator is trained only by the PI loss $\ell_{PI}^m$ and provides confidence mask following Equation 5, which does not touch noisy class labels for learning. During the inference stage, we discard the mask generator $f_m$ and only use the task executor for evaluation, which does not affect the inference speed.

Practically, the learning procedure relies heavily on the quality of the uncertainty estimation by $f_m$. Therefore, we pretrain the mask generator $f_m$ for $K$ epochs and re-weight the PI loss of the task executor $\ell_{PI}^t$ by Equation 6. Note that the task executor is still optimized by the un-reweighted PI loss and the noisy node classification loss during pretraining the mask generator. The training outline is presented in Algorithm 1.

## 5 EXPERIMENTS AND RESULTS

In this section, we present empirical evidence to validate the effectiveness of PI-GNN on different datasets with different noise types and ratios.

### 5.1 EXPERIMENTAL SETTING

**Datasets.** We used five datasets to evaluate PI-GNN, including Cora, CiteSeer and PubMed with the default dataset split as in (Kipf & Welling, 2017) and DBLP (Pan et al., 2016) as well as WikiCS dataset (Mernyei & Cangea, 2020). For the latter two datasets, we used the first 20 nodes from each class for training and the next 20 nodes for validation. The remaining nodes for each class are used as the test set. *The statistics of these datasets are summarized in Appendix Section B.*

Since all datasets are clean, following Patrini et al. (2017), we corrupted these datasets manually by the noise transition matrix $Q_{ij} = \Pr(\bar{y} = j \mid y = i)$ given that noisy $\bar{y}$ is flipped from clean $y$. Assume the matrix $Q$ has two representative structures: (1) Symmetry flipping (van Rooyen et al., 2015); (2) Asymmetric pair flipping: a simulation of fine-grained classification with noisy labels, where labelers may make mistakes only within very similar classes. Note the asymmetric case is much harder than the symmetry case. *Their precise definition is in Appendix Section A.*

We tested four different noise rates $\varepsilon \in \{0.2, 0.4, 0.6, 0.8\}$ in this paper for two different noise types, which cover lightly and extremely noisy supervision. Note that in the most extreme case, the noise rate 80% for pair flipping means 80% training data have wrong labels that cannot be learned without additional assumptions.

**Implementation details.** We used three different GNN architectures for evaluation, i.e., GCN, GAT and GraphSAGE, which are implemented following the package `torch-geometric` (Fey & Lenssen, 2019). All of them have two layers. Specifically, the hidden dimension of GCN, GAT and GraphSAGE is set to 16, 8 and 64, respectively. GAT has 8 heads for attention calculation in the first layer and 1 head in the second layer. The mean aggregator is used for GraphSAGE. We applied Adam optimizer (Kingma & Ba, 2015) with a learning rate of 0.01 for GCN and GraphSAGE and 0.005 for GAT. The weight decay is set to 5e-4. We conducted training for 400 epochs on a Tesla P40. The regularization loss weight $\beta$ is set to $|V|^2/(|V|^2 - M)^2$, where $|V|$ is the number of nodes and $M$ is the number of edges in the input graph $G$. The number of pretraining epochs $K$ is set to 50 and the total epoch $K$ is set to 400. The PI label is constructed based on the adjacency matrix by default. We tuned all the hyperparameters on the validation set and reported the node classification accuracy on the clean test set. *Details about the ablation studies on these factors are shown in Section 5.5.* Each experiment is repeated for 10 times with random seeds from 1 to 10.

### 5.2 EFFECTIVENESS ON DIFFERENT DATASETS

We evaluated the effectiveness of PI-GNN on five datasets with different noisy labels and noise rates, which is shown in Table 1 with GCN as the backbone. Specifically, we are interested to observe 1) whether the introduced pairwise interactions between nodes can improve a vanilla GNN against noisy

Table 1: Test accuracy on five different datasets for PI-GNN with GCN as the backbone. **Bold** numbers are superior results. Standard deviation is shown in the bracket.

| Noise type | No Noise | Symmetric Noise | | | | Asymmetric Noise | | | |
|---|---|---|---|---|---|---|---|---|---|
| Noise ratio | 0.0 | 0.2 | 0.4 | 0.6 | 0.8 | 0.2 | 0.4 | 0.6 | 0.8 |
| *Cora* | | | | | | | | | |
| GCN | **0.804(0.01)** | 0.722(0.03) | 0.613(0.07) | 0.446(0.06) | 0.285(0.07) | 0.703(0.04) | 0.514(0.06) | 0.291(0.04) | 0.161(0.02) |
| PI-GNN wo/ ue | 0.781(0.01) | **0.738(0.02)** | 0.654(0.05) | 0.510(0.04) | 0.287(0.06) | 0.717(0.04) | 0.563(0.07) | **0.349(0.06)** | **0.232(0.06)** |
| PI-GNN | 0.780(0.01) | 0.732(0.02) | **0.664(0.03)** | **0.515(0.03)** | **0.296(0.05)** | **0.723(0.03)** | **0.587(0.07)** | 0.347(0.07) | 0.209(0.06) |
| *CiteSeer* | | | | | | | | | |
| GCN | 0.683(0.01) | 0.603(0.02) | 0.524(0.04) | 0.382(0.04) | 0.230(0.03) | 0.595(0.03) | 0.465(0.05) | 0.281(0.05) | 0.171(0.05) |
| PI-GNN wo/ ue | 0.656(0.03) | 0.606(0.03) | 0.526(0.05) | 0.378(0.05) | 0.227(0.04) | 0.588(0.04) | 0.472(0.05) | 0.328(0.03) | 0.235(0.03) |
| PI-GNN | **0.684(0.03)** | **0.642(0.03)** | **0.591(0.03)** | **0.432(0.07)** | **0.245(0.05)** | **0.628(0.03)** | **0.531(0.06)** | **0.353(0.06)** | **0.238(0.06)** |
| *PubMed* | | | | | | | | | |
| GCN | **0.786(0.01)** | 0.707(0.02) | 0.610(0.06) | 0.462(0.07) | 0.367(0.07) | 0.682(0.05) | 0.524(0.08) | 0.399(0.06) | 0.387(0.07) |
| PI-GNN wo/ ue | 0.774(0.00) | 0.723(0.03) | 0.628(0.05) | 0.458(0.07) | **0.374(0.06)** | 0.722(0.03) | 0.579(0.07) | 0.412(0.05) | **0.405(0.03)** |
| PI-GNN | 0.774(0.00) | **0.724(0.03)** | **0.638(0.04)** | **0.470(0.08)** | 0.370(0.07) | **0.723(0.03)** | **0.583(0.07)** | **0.425(0.07)** | 0.401(0.04) |
| *DBLP* | | | | | | | | | |
| GCN | **0.641(0.02)** | 0.542(0.09) | 0.448(0.08) | 0.266(0.04) | 0.246(0.06) | 0.503(0.10) | 0.376(0.08) | 0.284(0.09) | 0.204(0.08) |
| PI-GNN wo/ ue | 0.622(0.05) | **0.565(0.12)** | 0.455(0.12) | 0.294(0.08) | 0.253(0.09) | 0.521(0.08) | 0.399(0.06) | **0.334(0.09)** | **0.291(0.12)** |
| PI-GNN | 0.635(0.04) | 0.564(0.13) | **0.456(0.10)** | **0.301(0.08)** | **0.258(0.11)** | **0.558(0.08)** | **0.453(0.09)** | 0.327(0.11) | 0.261(0.14) |
| *WikiCS* | | | | | | | | | |
| GCN | **0.703(0.01)** | 0.635(0.03) | 0.558(0.04) | 0.376(0.05) | 0.183(0.05) | 0.608(0.05) | 0.468(0.05) | 0.272(0.05) | 0.129(0.07) |
| PI-GNN wo/ ue | 0.676(0.01) | 0.624(0.02) | 0.552(0.05) | 0.396(0.07) | 0.197(0.07) | 0.607(0.03) | **0.483(0.05)** | **0.303(0.05)** | 0.125(0.05) |
| PI-GNN | 0.676(0.01) | **0.636(0.02)** | **0.562(0.04)** | **0.398(0.07)** | **0.208(0.07)** | **0.610(0.04)** | 0.479(0.05) | 0.300(0.04) | **0.135(0.06)** |

labels and 2) whether the uncertainty estimation to combat the sub-optimal PI labels is beneficial for the test set accuracy. Therefore, we compared the accuracy of a vanilla GNN, PI-GNN trained without the mask generator for uncertainty estimation (PI-GNN wo/ ue) and PI-GNN.

From Table 1, we made several observations: **Firstly**, the GNN trained with the PI learning objective is more robust to noisy labels, where both PI-GNN wo/ ue and PI-GNN perform much better than a vanilla GNN. **Secondly**, by performing uncertainty estimation by an additional mask generator and weighting the PI loss with the confidence mask, the negative effect of noisy labels is further reduced. For instance, PI-GNN improves the accuracy by 1.1% with the symmetric noise (noise ratio $\varepsilon = 0.8$) on Cora compared to PI-GNN wo/ ue, which justifies the effectiveness of our design. **Thirdly**, the PI learning objective does not help the GNN with the clean node labels, e.g., 80.4% of a vanilla GCN vs. 78.0% of PI-GNN on Cora, which illustrates the PI-GNN helps to combat noisy supervision rather than inherently improve the node classification with purely clean node labels. *Additional results on heterophilous datasets and with lower noise ratios are presented in Appendix Sections D and G.*

Table 2: Test accuracy on different graph neural network architectures. **Bold** numbers are superior results. Standard deviation is shown in the bracket.

| Noise type | No Noise | Symmetric Noise | | | | Asymmetric Noise | | | |
|---|---|---|---|---|---|---|---|---|---|
| Noise ratio | 0.0 | 0.2 | 0.4 | 0.6 | 0.8 | 0.2 | 0.4 | 0.6 | 0.8 |
| *Cora* | | | | | | | | | |
| GAT | **0.813(0.01)** | 0.741(0.03) | 0.647(0.07) | 0.474(0.06) | 0.273(0.06) | 0.714(0.04) | 0.516(0.07) | 0.288(0.05) | 0.172(0.05) |
| PI-GNN wo/ ue | 0.780(0.01) | 0.743(0.02) | 0.690(0.05) | **0.517(0.07)** | 0.261(0.04) | **0.730(0.03)** | **0.574(0.07)** | **0.330(0.05)** | **0.206(0.05)** |
| PI-GNN | 0.790(0.01) | **0.746(0.03)** | **0.691(0.05)** | 0.516(0.05) | **0.274(0.03)** | 0.728(0.03) | 0.569(0.05) | 0.329(0.05) | 0.192(0.05) |
| GraphSAGE | **0.805(0.01)** | 0.722(0.02) | 0.611(0.04) | 0.429(0.06) | 0.280(0.07) | 0.704(0.04) | 0.517(0.05) | 0.296(0.05) | 0.162(0.04) |
| PI-GNN wo/ ue | 0.775(0.03) | 0.735(0.03) | 0.666(0.03) | 0.502(0.06) | 0.298(0.07) | 0.715(0.03) | 0.581(0.08) | 0.368(0.08) | **0.241(0.06)** |
| PI-GNN | 0.786(0.01) | **0.756(0.01)** | **0.721(0.02)** | **0.584(0.06)** | **0.308(0.07)** | **0.755(0.02)** | **0.640(0.08)** | **0.393(0.09)** | 0.239(0.06) |
| *CiteSeer* | | | | | | | | | |
| GAT | **0.681(0.01)** | 0.614(0.03) | 0.542(0.03) | 0.394(0.05) | 0.234(0.05) | 0.588(0.04) | 0.451(0.06) | 0.269(0.04) | 0.175(0.05) |
| PI-GNN wo/ ue | 0.668(0.01) | **0.618(0.02)** | 0.544(0.04) | 0.384(0.06) | 0.228(0.04) | **0.599(0.03)** | 0.475(0.05) | 0.325(0.03) | **0.221(0.04)** |
| PI-GNN | 0.668(0.01) | 0.616(0.03) | **0.549(0.04)** | **0.398(0.06)** | **0.237(0.04)** | 0.597(0.03) | **0.485(0.05)** | **0.330(0.04)** | 0.214(0.04) |
| GraphSAGE | **0.698(0.01)** | 0.614(0.03) | 0.539(0.03) | 0.396(0.03) | 0.247(0.02) | 0.605(0.03) | 0.475(0.05) | 0.294(0.04) | 0.177(0.04) |
| PI-GNN wo/ ue | 0.667(0.01) | 0.609(0.02) | 0.555(0.03) | 0.411(0.06) | 0.226(0.04) | 0.613(0.03) | 0.519(0.05) | 0.356(0.05) | 0.236(0.05) |
| PI-GNN | 0.693(0.01) | **0.674(0.02)** | **0.627(0.03)** | **0.503(0.08)** | **0.256(0.07)** | **0.669(0.02)** | **0.593(0.05)** | **0.376(0.08)** | **0.237(0.06)** |

## 5.3 PERFORMANCE ON DIFFERENT GNN ARCHITECTURES

We evaluated our proposed PI-GNN on different GNN architectures, i.e., GAT and GraphSAGE. The experiments are conducted on Cora and CiteSeer dataset, which are shown in Table 2. As can be observed, our proposed approach performs similarly on GAT and GraphSAGE compared to the results on GCN, where the regularization of the pairwise interactions and the uncertainty estimation are both beneficial for model generalization even with extremely noisy supervision. Moreover, the uncertainty estimation is more effective on GraphSAGE. For example, in the Cora dataset, PI-GNN improves PI-GNN wo/ ue by 4.2% and 3.1% on average under symmetric noise and asymmetric noise, respectively, which is larger than that for GAT and GCN. It may suggest the mean aggregator in GraphSAGE is more susceptible to the sub-optimal PI labels. *We provide significance test on the results of GAT in Appendix Section E.*

## 5.4 COMPARISON WITH BASELINES

In order to further demonstrate the competitive performance of PI-GNN, we compared with several powerful baselines for combating noisy labels in literature. For a fair comparison, we used the same

Table 3: Comparative results with baselines. **Bold** numbers are superior results. LPM-1 means one extra clean label is used for each class. The result on the left and right of each cell is the classification accuracy of the Cora dataset and CiteSeer dataset, respectively.

| Noise type | Symmetric Noise | | Asymmetric Noise | |
|---|---|---|---|---|
| Noise ratio | 0.4 | 0.6 | 0.2 | 0.4 |
| | Test dataset: Cora / CiteSeer | | | |
| Decoupling | 0.581(0.06) / 0.518(0.03) | 0.425(0.06) / 0.390(0.03) | 0.696(0.03) / 0.581(0.03) | 0.541(0.05) / 0.474(0.04) |
| GCE | 0.627(0.07) / 0.530(0.03) | 0.447(0.06) / 0.383(0.03) | 0.710(0.04) / 0.598(0.03) | 0.511(0.05) / 0.468(0.05) |
| APL | 0.624(0.08) / 0.522(0.04) | 0.446(0.06) / 0.376(0.04) | 0.707(0.05) / 0.580(0.04) | 0.507(0.06) / 0.456(0.06) |
| Co-teaching | 0.577(0.11) / 0.573(0.07) | 0.376(0.07) / 0.404(0.06) | 0.706(0.06) / 0.616(0.04) | 0.457(0.10) / 0.462(0.08) |
| LPM-1 | 0.542(0.09) / 0.467(0.06) | 0.447(0.07) / 0.395(0.08) | 0.674(0.09) / 0.563(0.09) | 0.481(0.07) / 0.506(0.08) |
| T-Revision | 0.596(0.06) / 0.518(0.03) | 0.425(0.06) / 0.380(0.04) | 0.693(0.04) / 0.591(0.04) | 0.512(0.06) / 0.457(0.06) |
| DivideMix | 0.628(0.06) / 0.515(0.05) | 0.463(0.09) / 0.355(0.05) | 0.646(0.01) / 0.498(0.01) | 0.428(0.01) / 0.396(0.03) |
| PI-GNN | **0.664(0.03) / 0.591(0.03)** | **0.515(0.03) / 0.432(0.07)** | **0.723(0.03) / 0.628(0.03)** | **0.587(0.07) / 0.531(0.06)** |

GNN architecture, i.e., GCN, and the same overlapping hyperparameters during implementation. The other method-specific hyperparameters are tuned according to the original paper on the validation set. Specifically, we compared with noise-transition matrix-based method, T-revision (Xia et al., 2019), robust loss functions, such as Generalized Cross Entropy (GCE) loss (Zhang & Sabuncu, 2018) and Active Passive Loss (APL) (Ma et al., 2020), optimization-based approaches, such as Co-teaching (Han et al., 2018b), Decoupling (Malach & Shalev-Shwartz, 2017) and DivideMix (Li et al., 2020b). We also compared with Label Propagation and Meta learning (LPM) (Xia et al., 2021), a method that is specifically designed for solving label noise for node classification but uses a small set of clean nodes for assistance. We reported the classification accuracy on Cora and CiteSeer with symmetric noise (noise rate $\varepsilon = 0.4, 0.6$) and asymmetric noise (noise rate $\varepsilon = 0.2, 0.4$) in Table 3.

From Table 3, PI-GNN outperforms different baselines with a considerable margin, especially under extremely noisy supervisions, e.g., improving the classification accuracy by 3.7% on CiteSeer under the symmetric noise (noise rate $\varepsilon = 0.6$). Moreover, PI-GNN is able to outperform LPM-1, which relieves the strong assumption that auxiliary clean node labels are used for training. *We present comparison with traditional graph semi-supervised learning approaches in Appendix Section C.*

## 5.5 ABLATION STUDIES

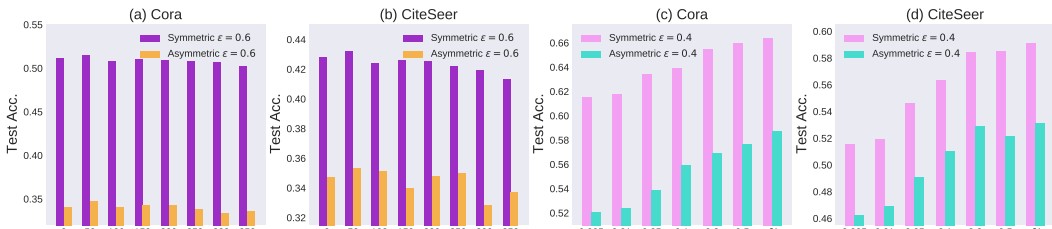

Figure 3: (a)-(b) Performance of PI-GNN *w.r.t.* different pretraining epochs on Cora and CiteSeer. x axis denotes the value of the pretraining epochs for the mask generator. (c)-(d) Performance of PI-GNN *w.r.t.* the regularization loss weight $\beta$. x axis denotes the value of the loss weight and $\beta'$ is the weight that is aware of the sparsity of the input graph.

**Sensitivity to the pretraining epoch of the mask generator.** We investigated whether the performance of PI-GNN is sensitive to the number of pretraining epochs for the mask generator. The experimental results on Cora and CiteSeer with GCN under symmetric and asymmetric noise (noise rate $\varepsilon = 0.6$) are shown in Figure 3 (a) and (b). As can be observed, pretraining the mask generator for $K$ epochs is effective for improving the generalization on the clean test set. Given a small $K$, the confidence mask is not estimated well which is not helpful to apply it on the task executor for regularization. Meanwhile, $K$ should not be too large in order to sufficiently regularize the task executor by the uncertainty-aware training objective. $K$ is set to 50 for all the experiments.

**The effect of regularization weight.** To observe whether the regularization loss weight $\beta$ matters to the model performance, we trained PI-GNN with different values of $\beta$, i.e., 0.005, 0.01, 0.05, 0.1, 0.2, 0.5 and compared with the value $\beta' = |V|^2/(|V|^2 - M)^2$ which is aware of the sparsity of the graph in Figure 3 (c) and (d). We conducted experiments on Cora and CiteSeer with GCN and showed the results with symmetric and asymmetric noise (noise rate $\varepsilon = 0.4$). From the figure, PI-GNN is sensitive to the choice of regularization loss weight $\beta$. On both datasets with different noise types, PI-GNN trained with $\beta'$ achieves the best clean test accuracy, and simultaneously avoids heavy tuning procedure on the validation set.

Table 5: **Left:** Performance of the PI-GNN applied on different label-noise baselines on Cora. **Right:** Performance of PI-GNN with different architectures for two branches on CiteSeer.

| Noise Type | Sym. Noise | Asym. Noise |
|---|---|---|
| Noise Ratio | 0.4 | 0.4 |
| APL Ma et al. (2020) | 0.624(0.08) | 0.507(0.06) |
| APL+PI-GNN | **0.656(0.05)** | **0.549**(0.07) |
| T-revision Xia et al. (2019) | 0.596(0.06) | 0.512(0.06) |
| T-revision+PI-GNN | **0.615(0.04)** | **0.536**(0.05) |
| DivideMix Li et al. (2020b) | 0.628(0.06) | 0.428(0.01) |
| DivideMix+PI-GNN | **0.674(0.03)** | **0.442**(0.02) |

| Noise Type | Sym. Noise | Asym. Noise |
|---|---|---|
| Noise Ratio | 0.8 | 0.6 |
| GAT only | 0.237(0.04) | 0.330(0.04) |
| GCN-GAT | **0.241(0.05)** | **0.332(0.03)** |
| GraphSAGE only | 0.256(0.07) | 0.376(0.08) |
| GCN-GraphSAGE | **0.266(0.05)** | **0.383(0.06)** |
| GraphSAGE only | 0.256(0.07) | 0.376(0.08) |
| GAT-GraphSAGE | **0.271(0.04)** | **0.381(0.06)** |

**The effect of pairwise interaction labels.** To demonstrate the importance of informative PI labels for PI-GNN, we tested PI-GNN under different PI labels in addition to those based on adjacency matrix, namely by 1) noisy class label comparison and 2) random labels. We also compared with using the task executor $f_t$ to generate the confidence mask without introducing the mask generator. We used GCN as the backbone and tested on two datasets, i.e., Cora and CiteSeer and different noise types. The results are shown in Table 4. From Table 4, using PI labels based on the graph structure obtains the best performance. Directly comparing the similarity of the noisy class labels to get the PI labels or using randomly generated PI labels incurred the worst performance because of the heavy noise during the PI label generation. Moreover, removing the mask generator for confidence

Table 4: Performance of PI-GNN with different PI labels.

| Noise Type | Symmetric Noise | Asymmetric Noise |
|---|---|---|
| | Cora | |
| Noise Ratio | 0.6 | 0.6 |
| Noisy label comparison | 0.453(0.05) | 0.289(0.04) |
| Random PI label | 0.449(0.05) | 0.295(0.04) |
| Task executor only | 0.511(0.03) | 0.329(0.05) |
| Adjacency matrix | **0.515(0.03)** | **0.347(0.07)** |
| | CiteSeer | |
| Noise Ratio | 0.6 | 0.6 |
| Noisy label comparison | 0.339(0.04) | 0.279(0.05) |
| Random PI label | 0.351(0.04) | 0.298(0.04) |
| Task executor only | 0.430(0.07) | 0.340(0.05) |
| Adjacency matrix | **0.432(0.07)** | **0.353(0.06)** |

mask generation decreases the test accuracy because the node embeddings are optimized by the noisy class labels as well, which is ineffective for the task executor to generalize even with uncertainty-aware training by itself (32.9% vs. 34.7% for Cora with asymmetric noise and a noise rate of 60%). *We also compare with training with the clean PI labels, which can be regarded as an upper bound of our method in Appendix Section F.*

**Application of PI-GNN on label-noise baselines.** To observe whether PI-GNN is able to improve the generalization ability for different label-noise baseline models, we extended three representative approaches, i.e., T-revision (Xia et al., 2019), APL (Ma et al., 2020) and DivideMix (Li et al., 2020b) by adding the PI learning objective during training. Specifically, we used the sum of the original loss and the PI loss to optimize the GNN. The weight for PI loss is set to $\beta'$. We chose GCN as the backbone and reported the test accuracy on Cora with both symmetric noise (noise rate $\varepsilon = 0.4$) and asymmetric noise (noise rate $\varepsilon = 0.4$) in Table 5 **Left**. As the result shows, PI-GNN is orthogonal to those robust baseline models, which is potentially useful for improving their performance without bells and whistles. For instance, The test set accuracy is improved by 4.6% on DivideMix under the symmetric noise and thus demonstrates the universality of our proposed PI-GNN.

**Different architectures for two branches.** PI-GNN allows for a flexible choice of the architectures for the mask generator and the task executor, where a light-weight mask generator can help a large task executor for node classification during the uncertainty-aware training. In what follows, we used three different mask generator-task executor pairs, namely GCN-GAT, GCN-GraphSAGE and GAT-GraphSAGE. The number of parameters for GCN, GAT, GraphSAGE is 0.02, 0.09 and 0.18 M, respectively. The comparison with using the same architectures are shown in Table 5 **Right**. From Table 5 **Right**, using a light-weight GNN for the mask generator is able to further improve the clean test accuracy, which is promising for efficient deployment of PI-GNN on real-world graph datasets.

## 6 CONCLUSION

In this paper, we proposed PI-GNN, a simple but effective learning paradigm for helping the graph neural networks to generalize well with noisy supervision. Our key idea is to leverage the pairwise interactions between nodes to explicitly adjust the similarity of those node embeddings during training. In order to alleviate the negative effect of the collected sub-optimal PI labels, we introduce a new uncertainty-aware training approach to reweight the PI learning objective by its prediction confidence. We conducted extensive experiments to demonstrate that PI-GNN can train GNNs robustly under extremely noisy supervision, which serves as a crucial step towards the reliable deployment of GNNs in complex real-world applications.

ETHICS STATEMENT

This paper doesn't raise any ethics concerns. This study doesn't involve any human subjects, practices to data set releases, potentially harmful insights, methodologies and applications, potentially conflicts of interest and sponsorship, discrimination/bias/fairness concerns, privacy and security issues, legal compliance, and research integrity issues.

REPRODUCIBILITY STATEMENT

To ensure the reproducibility of experimental results, we will provide source codes of this paper using an anonymous repository link in the discussion phase.

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

# Supplementary Material

## A  DEFINITION OF NOISE

The definition of transition matrix $Q$ is as follows. $n$ is number of the class.

Asymmetric pair flipping:

$$
Q = \begin{bmatrix}
1-\varepsilon & \varepsilon & 0 & \ldots & 0 \\
0 & 1-\varepsilon & \varepsilon & & 0 \\
\vdots & & \ddots & \ddots & \vdots \\
0 & & & 1-\varepsilon & \varepsilon \\
\varepsilon & 0 & \ldots & 0 & 1-\varepsilon
\end{bmatrix},
\tag{8}
$$

Symmetry flipping:

$$
Q = \begin{bmatrix}
1-\varepsilon & \frac{\varepsilon}{n-1} & \ldots & \frac{\varepsilon}{n-1} & \frac{\varepsilon}{n-1} \\
\frac{\varepsilon}{n-1} & 1-\varepsilon & \frac{\varepsilon}{n-1} & \ldots & \frac{\varepsilon}{n-1} \\
\vdots & & \ddots & & \vdots \\
\frac{\varepsilon}{n-1} & \ldots & \frac{\varepsilon}{n-1} & 1-\varepsilon & \frac{\varepsilon}{n-1} \\
\frac{\varepsilon}{n-1} & \frac{\varepsilon}{n-1} & \ldots & \frac{\varepsilon}{n-1} & 1-\varepsilon
\end{bmatrix}.
\tag{9}
$$

## B  DATASET DETAILS

Here we provide the details of graph datasets for node classification.

Table 6: Statistics of the datasets.

| Dataset | #Nodes | #Edges | #Classes |
|---------|--------|--------|----------|
| Cora | 2,485 | 5,069 | 7 |
| CiteSeer | 2,110 | 3,668 | 6 |
| PubMed | 19,717 | 44,324 | 3 |
| DBLP | 17,716 | 105,734 | 4 |
| WikiCS | 11,701 | 216,123 | 10 |

## C  COMPARISON WITH TRADITIONAL GRAPH SEMI-SUPERVISED LEARNING BASED APPROACHES.

For the comparison with the traditional semi-supervised graph embedding methods, we follow the same experimental setting and compare with ICA (Lu & Getoor, 2003), Planetoid (Yang et al., 2016) and Label Propagation (LP) (Zhu & Ghahramani, 2002) on Cora as follows. The result shows the advantage of PI-GNN across different noise ratios.

Table 7: Comparison with more baselines on Cora Dataset.

| Noise type | Symmetric Noise | | | | | Asymmetric Noise | | | |
|------------|------|------|------|------|------|------|------|------|------|
| Noise ratio | 0.0 | 0.2 | 0.4 | 0.6 | 0.8 | 0.2 | 0.4 | 0.6 | 0.8 |
| ICA | 0.729(0.01) | 0.609(0.01) | 0.523(0.04) | 0.394(0.00) | 0.159(0.00) | 0.549(0.00) | 0.453(0.00) | 0.284(0.01) | 0.127(0.01) |
| LP | 0.603(0.00) | 0.506(0.02) | 0.417(0.03) | 0.297(0.03) | 0.170(0.03) | 0.513(0.03) | 0.391(0.04) | 0.238(0.03) | 0.141(0.02) |
| Planetoid | 0.739(0.01) | 0.639(0.03) | 0.527(0.04) | 0.379(0.05) | 0.265(0.06) | 0.627(0.03) | 0.441(0.04) | 0.271(0.06) | 0.210(0.09) |
| PI-GNN | **0.780(0.01)** | **0.732(0.02)** | **0.664(0.03)** | **0.515(0.03)** | **0.296(0.05)** | **0.723(0.03)** | **0.587(0.07)** | **0.347(0.07)** | **0.211(0.06)** |

## D  EXPERIMENTAL RESULTS ON HETEROPHILOUS DATASETS

We perform extra experiments on heterophilous datasets (Ma et al., 2021). The results are in the following table. It shows that PI-GNN is still able to outperform the vanilla one except for one case

in Chameleon dataset. Meanwhile, the improvement is somewhat smaller, which implies PI-GNN may be more effective on homophilous datasets.

Table 8: Experimental results on heterophilous datasets.

| Noise type | Symmetric | | Asymmetric | |
|---|---|---|---|---|
| | Actor | | | |
| Noise Ratio | 0.4 | 0.6 | 0.4 | 0.6 |
| GCN | 0.209(0.02) | 0.208(0.02) | 0.198(0.03) | 0.199(0.02) |
| PI-GNN w/o ue | 0.216(0.01) | 0.210(0.02) | 0.201(0.02) | **0.202(0.02)** |
| PI-GNN | **0.218(0.02)** | **0.213(0.02)** | **0.204(0.02)** | 0.200(0.02) |
| | Chameleon | | | |
| GCN | 0.251(0.03) | 0.246(0.03) | **0.245(0.04)** | 0.228(0.03) |
| PI-GNN w/o ue | 0.264(0.03) | 0.249(0.03) | 0.242(0.05) | 0.229(0.04) |
| PI-GNN | **0.269(0.02)** | **0.251(0.03)** | 0.239(0.05) | **0.237(0.04)** |

## E  SIGNIFICANCE TEST RESULTS

We perform significance tests to verify whether PI-GNN outperforms the vanilla GNN model significantly using double-sided T-test. We use python package "scipy.stats.ttest1samp" and report the average results over 10 different runs as follows. PI-GNN is better than GAT because the absolute value of the t-statistic is relatively large and the p-value is small.

Table 9: Statistical significance tests.

| Method | Setting | T-statistic | p-value |
|---|---|---|---|
| | Cora | | |
| GAT vs. PI-GNN | Symmetric Noise-0.8 | 4.78 | 0.001 |
| | Asymmetric Noise-0.8 | 3.42 | 0.008 |
| | CiteSeer | | |
| GAT vs. PI-GNN | Symmetric Noise-0.8 | 2.09 | 0.060 |
| | Asymmetric Noise-0.8 | 4.63 | 0.001 |

## F  EXPERIMENTAL RESULTS ON USING CLEAN PI LABELS

To observe the node classification results with clean PI labels, we did experiments on Cora, CiteSeer and PubMed with GCN and a noise ratio of 0.4 and 0.6. The results are shown in the Table 10. In most cases, clean PI labels can help the PI-GNN to combat noisy labels except for some challenging cases with asymmetric noise. One reason may be the inherent noise exists in clean node labels for Cora, where we cannot obtain perfectly clean PI label.

Table 10: Experimental results on using clean PI labels. Clean PI-GNN means the PI-GNN is trained with the clean PI labels.

| Noise type | Symmetric | | Asymmetric | |
|---|---|---|---|---|
| | Cora | | | |
| Noise Ratio | 0.4 | 0.6 | 0.4 | 0.6 |
| PI-GNN | 0.664(0.03) | 0.515(0.03) | 0.587(0.07) | **0.347(0.07)** |
| Clean PI-GNN | **0.671(0.03)** | **0.523(0.03)** | **0.589(0.07)** | 0.341(0.07) |
| | CiteSeer | | | |
| PI-GNN | 0.591(0.03) | 0.432(0.07) | 0.531(0.06) | 0.353(0.06) |
| Clean PI-GNN | **0.605(0.04)** | **0.447(0.07)** | **0.536(0.05)** | **0.355(0.05)** |
| | PubMed | | | |
| PI-GNN | 0.638(0.04) | 0.470(0.08) | 0.583(0.07) | 0.425(0.07) |
| Clean PI-GNN | **0.640(0.02)** | **0.485(0.07)** | **0.590(0.07)** | **0.429(0.07)** |

## G  EXPERIMENTAL RESULTS WITH LOWER NOISE RATIOS

For the PI-GNN under lower noise ratios, we empirically verify its effectiveness on Cora and CiteSeer with the noise ratio of 0.1, which is shown in the following table.

Table 11: Experimental results with lower noise ratios.

| Noise type | Symmetric | Asymmetric |
|---|---|---|
| | Cora | |
| Noise Ratio | 0.1 | 0.1 |
| GCN | 0.766(0.03) | 0.762(0.04) |
| PI-GNN wo/ ue | 0.769(0.03) | 0.763(0.03) |
| PI-GNN | **0.772(0.02)** | **0.768(0.03)** |
| | CiteSeer | |
| Noise Ratio | 0.1 | 0.1 |
| GCN | 0.642(0.03) | 0.618(0.05) |
| PI-GNN wo/ ue | 0.648(0.04) | 0.633(0.02) |
| PI-GNN | **0.659(0.03)** | **0.658(0.05)** |

