# OpenReview forum: "PI-GNN: Towards Robust Semi-Supervised Node Classification against Noisy Labels"
_ICLR.cc/2022/Conference — ICLR 2022 Submitted_

### Official Review · Reviewer_8yHP · 2021-10-28

**Correctness:** 3
**Technical Novelty And Significance:** 2
**Empirical Novelty And Significance:** 2
**Recommendation:** 3
**Confidence:** 4

**Details Of Ethics Concerns:**

No ethical concerns are raised.


**Main Review:**

This paper tackles an important problem in the graph representation learning community, uses a simple but effective approach, and conducts experiments with various settings (especially with/against label-noise baselines). However, it has several weaknesses and should be revised for the following points:

First, related work on graph structure learning is not thoroughly surveyed. Using a task to predict graph structure (i.e., adjacency matrix or PI label 1) and label agreement (i.e., PI label 2) is proposed in many existing works. However, this paper does not discuss them and only mentions a few works such as [Stretcu, Otilia, et al.]. I would recommend authors to survey prior works about their methods. Plus, these prior works are not empirically compared by experiments; thus, the technical novelty of the model might be overclaimed.

- Jiang, Bo, et al. "Semi-supervised learning with graph learning-convolutional networks." Proceedings of the IEEE/CVF Conference on Computer Vision and Pattern Recognition. 2019.
- Yu, Donghan, et al. "Graph-revised convolutional network." Joint European Conference on Machine Learning and Knowledge Discovery in Databases. Springer, Cham, 2020.
- Zheng, Cheng, et al. "Robust graph representation learning via neural sparsification." International Conference on Machine Learning. PMLR, 2020.
- Jin, Wei, et al. "Graph structure learning for robust graph neural networks." Proceedings of the 26th ACM SIGKDD International Conference on Knowledge Discovery & Data Mining. 2020.
- Chen, Yu, Lingfei Wu, and Mohammed Zaki. "Iterative deep graph learning for graph neural networks: Better and robust node embeddings." Advances in Neural Information Processing Systems 33 (2020).
- Kim, Dongkwan, and Alice Oh. "How to find your friendly neighborhood: Graph attention design with self-supervision." International Conference on Learning Representations. 2020.
- Fatemi, Bahare, Layla El Asri, and Seyed Mehran Kazemi. "SLAPS: Self-Supervision Improves Structure Learning for Graph Neural Networks." arXiv preprint arXiv:2102.05034 (2021).
- Stretcu, Otilia, et al. "Graph Agreement Models for Semi-Supervised Learning." Advances in Neural Information Processing Systems 32 (2019): 8713-8723.

Second, the complexity and scalability of models are not discussed. The confidence mask requires $|V| \times |V|$ space and time complexity, and it would be numerically large if we want to deal with a gigantic graph. However, this paper does not analyze how the model parts cost and how large graphs can be used. Here are some quick solutions: (1) conducting experiments on relatively large graphs (e.g., OGB datasets), (2) analyzing complexity and comparing with baselines, and (3) measuring the memory consumption and training/inference wall-clock time.

Third, experimental settings and their demonstration are not consistent. For example, some tables only present a subset of noise ratio settings (e.g., 0.4/0.6, 0.2/0.4 in Table 3, 0.4/0.6 in Table 8). Significant test results are only reported on Cora and CiteSeer (considering that we usually use asterisks, a space-efficient way to represent statistical significance). Is there any specific reason or motivation for this configuration?

Lastly, here are some minor concerns and suggestions:
- The term 'uncertainty estimation' seems far from the well-known meaning in the deep learning (especially in the Bayesian DNN) community. Therefore, it can be misleading to many potential readers. Researchers are now differentiating the softmax output and the model uncertainty (i.e., A model can be uncertain in its predictions even with a high softmax output). For more details, see Gal, Yarin, and Zoubin Ghahramani. "Dropout as a Bayesian approximation: Representing model uncertainty in deep learning." international conference on machine learning. PMLR, 2016.
- It would be great if the authors used line charts to show the performance against noise ratio of models. Tables are also good, but sometimes it is hard to recognize the difference and see the big picture.
- It would be great if the authors present the exact value of $\beta$ in Figure 3.


**Summary Of The Paper:**

This paper proposes the PI-GNN (Pairwise Interactions in Graph Neural Network) model for semi-supervised tasks for nodes with noisy labels. It employs a new auxiliary task to predict PI (pairwise interactions) labels using a pair of nodes. Since it is costly to get clean PI labels, the authors use (1) two instantiations of PI labels: adjacency matrix and class agreement after label propagation and (2) uncertainty-aware (pre-)training with two model branches. The experiments on five real-world benchmarks demonstrate that PI-GNN outperforms GNN models (GCN, GraphSAGE, and GAT) and methods for noisy label settings (Decoupling, GCE, DivideMix, and so on.) on various noise levels and settings. PI label learning and uncertainty-aware training both contribute to performance improvement.

**Summary Of The Review:**

Although this work has several strengths (important problem, simple but effective method, and extensive experiments), it is not ready for publications by following weaknesses:
- Related work on graph structure learning is not thoroughly surveyed and not compared. Technical novelty on the model might be overclaimed.
- Scalability is not discussed, even though it has the potential of huge complexity.
- Experimental settings and their demonstration are not consistent without justifications.

---

### Official Review · Reviewer_Lvq5 · 2021-11-02

**Correctness:** 3
**Technical Novelty And Significance:** 3
**Empirical Novelty And Significance:** 3
**Recommendation:** 5
**Confidence:** 4

**Main Review:**

1. From eqn(3), the labels of PI can be the input adjacency matrix A or are generated from label propagation. It is not clear how does label propagation generate PI labels? Is it a semi-supervised way? Are clean labels used? If so, in principle, this setting is incorrect for robustness research. If not, please give a more detailed explanation and training details. Besides, what does this sentence “which does not touch noisy class labels for learning” (on Page 6) mean?
2. How about the complexity, no analysis is given.
3. In experiments, although 5 datasets are evaluated in table1, only two small datasets (Cora and Citeseer) are used in all subsequent experiments. Both these two datasets belong to citation datasets, which have similar characteristics. DBLP and WikiCS datasets should be evaluated on ablation study.
4. From Fig.3 (c)-(d), the performance of PI-GNN always increases with \beta increasing. More \beta values should be used to test the trend of accuracy, the most extreme is how the performance when \beta=1.


**Summary Of The Paper:**

This paper is mainly dedicated to developing a robust model against noisy labels. Since it is observed from Fig. 1(a) that the noise ratio for the PI labels is lower than that of the point-wise labels, the authors introduce positive and negative sample pairs (Pair Interactions) to enhance model capabilities, which is likely contrastive learning. Moreover, the labels of positive and negative sample pairs may not be confident, e.g., sub-optimal PI labels with noises. To address this problem, the authors introduce another branch (Mask Generator) trained on pair interaction learning objective to generate a confidence map to reweight the loss of PI learning objective in Task Executor. Experimentally, the accuracies of the proposed method are higher than baselines in the case of noisy labels.

**Summary Of The Review:**

1. The presentation is overall good and easy to follow, but some descriptions or details are not clear.
2. The novelty is limited. Essentially, the proposed pair interactions belong to contrastive learning.
3. The experiments show better performance.

---

### Official Review · Reviewer_2qdw · 2021-11-02

**Correctness:** 4
**Technical Novelty And Significance:** 3
**Empirical Novelty And Significance:** 3
**Recommendation:** 5
**Confidence:** 3

**Main Review:**

The idea is the new approach is to learn Pairwise Intersection (PI) between graph nodes to enhance the robustness. Ideally, a node pair has positive PI is more likely to have the same labels. Authors further propose strategies to optimize such PI labels.

I think the motivation of this paper is good. One can expect to gain strong intersection between nodes with same label even in presence of noise and such structure upon the graph should have good impact on the robustness. What I am concerned is the experiments, I don not feel the empirical results in this paper achieve significantly accuracy upon vanilla GNN approach (like GCN). In table 1 and 2, some improvements are only from 0.273 to 0.274 or 0.367 to 0.370. Thus, I am not fully convinced the necessarily of using PI-GNN in similar tasks.

**Summary Of The Paper:**

This paper proposes a new GNN approach which they call PI-GNN that aims to improve performance in presence of noise or corruption. Empirical results suggest the new approach can slightly improve the accuracy when there is noise.

**Summary Of The Review:**

I give a borderline evaluation for this paper. My main complaint is the experiment results are not strong enough. More convincing results or a strong reason why PI-GNN is a good candidate in similar tasks should make me improve the evaluation.

---

### Official Review · Reviewer_bNFD · 2021-11-04

**Correctness:** 4
**Technical Novelty And Significance:** 2
**Empirical Novelty And Significance:** 3
**Recommendation:** 5
**Confidence:** 4

**Main Review:**

Strengths:

1. The problem is well-motivated.

This paper focuses on semi-supervised node classification in a setting where noisy node labels exist. As node classification is a fundamental problem in graph machine learning and noisy labels always exist due to some annotation errors, this problem looks well-motivated and interesting to me.

2. The idea of pairwise interactions is intuitive.

To deal with the problem, the authors propose a new objective called the pairwise interaction. Different from most training objectives which focus on pointwise interactions between nodes and labels, this objective forces similar nodes to have similar node embeddings, where the similarity is measured with both graph structures and node labels. Overall, this objective is intuitive in the context of node classification with noisy labels.

3. The model is simple yet effective.

Based on pairwise interactions, the authors propose PI-GNN, which consists of two GNNs, i.e., a mask generator and a task executor. The mask generator computes a soft mask matrix over node pairs, which is used to reweight node pairs for training. The framework is quite straightforward, and it is able to effectively model the pairwise interactions.

Weaknesses:

1. The novelty of the paper seems limited.

My major concern is on the novelty of the paper. In terms of problem, node classification with noisy labels has been explored quite extensively in literature. For example, as mentioned in related work, methods based on loss correction and sample reweighting have been applied to GNNs. Given these works, the problem studied in this paper is not very new.

Comparing this work with existing works, the authors claim that existing works only consider pointwise interactions while this paper focuses on pairwise interactions. However, the idea of modeling pairwise interactions is also not very new in the field of graph machine learning. For example, [1] uses a GNN to infer whether two linked nodes have the same label, and thus models pairwise interactions. [2] uses Gaussian process with a graph structure-based kernel for node label prediction, which can also achieve modeling pairwise interactions. [3] and [4] use CRFs to model pairwise interactions of node labels. Although all these works leverage pairwise interactions in the clean-label setting, they have very similar ideas to this paper, and they can be extended to deal with noisy labels quite easily.

Therefore, although this paper has some contributions over existing works, I think these contributions are small, and hence the novelty of the paper is limited.

2. Some model designs need more justification.

Although PI-GNN is very intuitive, some model designs need more justification.

In equation (5), why the mask matrix is designed in that way? Is there any example to motivate the design? It would be more convincing if the authors could further elaborate on that or provide some theoretical justification.

In PI-GNN, two GNNs are leveraged. To me, these two GNNs have quite similar roles in the sense that they both try to learn an embedding for each node. Is there any conceptual difference between the embeddings learned by the two GNNs? What if we use the same GNN encoder for both the mask generator and the task executor?

In equation (4), a pairwise loss function is used for training. According to the definition, the size of $B_{PI}^+$ is much larger than the size of $B_{PI}^+$, resulting in a problem of label imbalance. How does PI-GNN deal with the problem?

3. It is better to consider some larger datasets.

In the experiment, all the datasets are very small, and the largest one only has ~10K nodes. If I understand correctly, PI-GNN has higher memory cost than standard GNNs, as PI-GNN needs to compute a soft weight for each pair of nodes. Because of that, it is unclear whether PI-GNN can deal with very large graphs, and it would be more convincing if the authors could run such experiments.

[1] Stretcu, Otilia, Krishnamurthy Viswanathan, Dana Movshovitz-Attias, Emmanouil A. Platanios, Sujith Ravi, and Andrew Tomkins. "Graph Agreement Models for Semi-Supervised Learning." In NeurIPS. 2019.

[2] Ng, Yin Cheng, Nicolò Colombo, and Ricardo Silva. "Bayesian Semi-supervised Learning with Graph Gaussian Processes." In NeurIPS. 2018.

[3] Qu, Meng, Yoshua Bengio, and Jian Tang. "Gmnn: Graph markov neural networks." In ICML. 2019.

[4] Ma, Tengfei, Cao Xiao, Junyuan Shang, and Jimeng Sun. "CGNF: Conditional Graph Neural Fields." (2018).


**Summary Of The Paper:**

This paper focuses on semi-supervised node classification with noisy node labels. The authors propose a novel learning objective called the pairwise interactions, which encourages node pairs holding positive PI labels to have close node embeddings. Extensive experiments show promising results.

**Summary Of The Review:**

This paper focuses on semi-supervised node classification with label noise. Although the paper has some contributions over existing works, I think the contributions are not significant enough, so I lean towards a weak reject.

---

### Decision · Program_Chairs · 2022-01-20

**Decision:**

Reject

**Comment:**

The paper studies the noisy labels problem in semi-supervised node classification and proposes a method that leverages pairwise interactions to explicitly force the embeddings for certain node pairs to be close to each other leading to better robustness.

The reviewers agreed the proposed method is promising. However, the reviewers also had concerns about the novelty, and that certain aspects of the method could be justified better and the experiments should consider larger scale settings to make the paper more convincing. These were the key reasons for rejection.